Chronic fluoxetine treatment of juvenile zebrafish (Danio rerio) does not elicit changes in basal cortisol levels and anxiety-like behavior in adulthood

Petrunich-Rutherford Maureen L. mlpetrun@iun.edu
Department of Psychology, Indiana University Northwest , Gary, IN , USA
Piato Angelo
Electronic publication date: 2019 Mar 8
Publication date: 2019
Volume: 7
Electronic Location ID: e6407
Received 2018 Aug 22; Accepted 2019 Jan 8
Copyright: © 2019 Petrunich-Rutherford
Copyright year: 2019
Copyright holder: Petrunich-Rutherford
License: This is an open access article distributed under the terms of the Creative Commons Attribution License, which permits unrestricted use, distribution, reproduction and adaptation in any medium and for any purpose provided that it is properly attributed. For attribution, the original author(s), title, publication source (PeerJ) and either DOI or URL of the article must be cited.
License URL: https://creativecommons.org/licenses/by/4.0/

Keywords: Zebrafish, Juvenile, Cortisol, Anxiety, Fluoxetine, SSRI, Development, Stress, HPA, HPI

Funding: IU Northwest Faculty Grant-in-aid of Research, Project Initiation Grant, and Summer Faculty Fellowship This work was supported by the IU Northwest Faculty Grant-in-aid of Research, Project Initiation Grant, and Summer Faculty Fellowship. The funders had no role in study design, data collection and analysis, decision to publish, or preparation of the manuscript.

==============================
Exposure to selective serotonin reuptake inhibitors (SSRIs) during development may elicit long-term neuroadaptive changes that could alter the basal regulation of stress-associated physiological and behavioral processes later in life. Currently, the effects of juvenile fluoxetine exposure in rodent models appear to be dependent on the developmental window targeted as well as the duration of drug exposure. The zebrafish (Danio rerio) model is rapidly becoming a useful tool in pharmacological research and can be used to help elucidate some of the long-term effects of fluoxetine exposure prior to sexual maturation on neuroendocrine and behavioral stress markers. In the current study, juvenile zebrafish were chronically exposed to fluoxetine hydrochloride (0 or 100 μg/L) for 14 days (31–44 days post-fertilization (dpf)), then were left untreated until young adulthood. Starting at 90 dpf, basal neuroendocrine stress and behavioral responses of zebrafish were assessed. Cortisol was extracted from the young adult zebrafish body (trunk) and quantified via enzyme-linked immunosorbent assay (ELISA). Anxiety-like behaviors were assessed in response to introduction to the novel tank test. It was expected that juvenile exposure to fluoxetine would (1) reduce basal cortisol levels and (2) elicit anxiolytic effects in the novel tank test in adulthood. However, fluoxetine exposure during the juvenile period was not associated with alterations in basal levels of cortisol nor were there any significant changes in anxiety-like behavior in the young adult zebrafish. Thus, in zebrafish, it does not appear that SSRI exposure during the juvenile period has a long-term adverse or maladaptive impact on the basal expression of cortisol and anxiety-like behavior in adulthood. Further studies are needed to determine if SSRI exposure during this developmental window influences neuroendocrine and behavioral responses to acute stress.

Introduction

Adolescence is a developmental time marked by tremendous amounts of brain growth and plasticity. However, it is also a period of particular mental vulnerability. It has been estimated that 12.5% of the U.S. population aged 12–17—around three million adolescents—has had at least one episode of depression in the last year (Center for Behavioral Health Statistics and Quality, 2016). These depressive episodes can be isolated, or may be indicative of a chronic, relapsing condition that can be lifelong. Selective serotonin reuptake inhibitors (SSRIs) are currently the most effective pharmaceutical option for treating adolescent depressive and anxiety disorders (Ryan, 2003; Bostic et al., 2005; Gentile, 2010; Kodish, Rockhill & Varley, 2011; Hetrick et al., 2012). Indeed, the SSRI fluoxetine (Prozac) is the only FDA-approved drug for treating childhood and adolescent depression, although other SSRIs are often prescribed on an “off-label” basis (Food and Drug Administration (FDA), 2017). However, caution is still advised when prescribing antidepressants to youth, as the risk of increased suicidality or other negative side effects cannot be completely ruled out (Reeves & Ladner, 2010; Cousins & Goodyer, 2015; Vitiello & Ordóñez, 2016). Additionally, in rodent models, chronic SSRI treatment prior to sexual maturation can elicit unique and/or long-lasting effects that can persist until adulthood, such as changes in the brain’s serotonin-producing neurons (Maciag et al., 2006) and alterations in behavior (Ansorge et al., 2004; Oh et al., 2009; Iñiguez, Warren & Bolaños-Guzmán, 2010; Iniguez et al., 2014). These unpredictable and unexpected changes in brain structure or function may potentially preclude the use of serotonergic antidepressants in adulthood, should depression be a chronic, lifelong condition.

The zebrafish (Danio rerio) animal model is rapidly growing in popularity, and is a validated, low-cost, low-maintenance option for use in pharmacological and stress research due to the conservation of mechanisms regulating biological responses to stress (for reviews, see Steenbergen, Richardson & Champagne, 2011; Stewart et al., 2014). The neuroendocrine stress axis in teleost species (such as zebrafish), the hypothalamic-pituitary-interrenal (HPI) axis, is analogous to the hypothalamic-pituitary-adrenal (HPA) axis of mammals (Wendelaar Bonga, 1997; Nesan & Vijayan, 2013). As observed in rodents, SSRIs can also alter stress responses in zebrafish. For example, acute exposure to fluoxetine in adult zebrafish suppresses cortisol responses (De Abreu et al., 2014; Abreu et al., 2015, 2017), increases extracellular levels of brain serotonin (Maximino et al., 2014), and alters behavioral responses (Maximino et al., 2014; Giacomini et al., 2016a) observed in response to acute stress exposure. Paradoxically, however, acute fluoxetine exposure in the absence of environmental stressors is associated with increased cortisol responses and alterations in brain mRNA levels for stress-axis related transcripts (Theodoridi, Tsalafouta & Pavlidis, 2017). Additionally, acute fluoxetine exposure in the absence of stress has been found to elicit both anxiogenic and anxiolytic effects on behavior, depending on the type of test used (Maximino et al., 2013; Magno et al., 2015). Although acute fluoxetine exposure studies are important for elucidating the early neural and behavioral responses to drug exposure, SSRIs are typically used chronically; therefore, chronic treatment paradigms in non-human animals are essential for understanding long-term neuroadaptive or behavioral effects elicited in response to putative serotonergic alterations. Indeed, studies involving chronic exposure of fluoxetine show more consistent results compared to studies investigating acute exposure in zebrafish. Chronic fluoxetine treatment of adult zebrafish induces anxiolytic effects in behavioral tests (Maximino et al., 2011; Wong, Oxendine & Godwin, 2013), alters whole-brain gene expression of some stress-related genes (Wong, Oxendine & Godwin, 2013), blunts chronic stress-induced increases in cortisol levels (Giacomini et al., 2016b; Marcon et al., 2016), and attenuates chronic stress-induced anxiogenic behaviors and the whole-brain expression of gene markers of inflammation (Marcon et al., 2016).

The effects of fluoxetine are likely dependent on a host of factors, including the time and duration of drug exposure, whether the dependent measures are assessed after a stressor or not, etc. Developmental stage is also crucial for understanding the impact of serotonergic modulation on long-term outcomes (Kiryanova, McAllister & Dyck, 2013), including the development and expression of HPA/HPI neuroendocrine responses and anxiety-like behaviors. Exposure to fluoxetine during the embryonic stage affects mortality and growth (Kalichak et al., 2016). Additionally, fluoxetine exposure during the embryonic and larval periods alters gene expression (Park et al., 2012; Cunha et al., 2016; Crago & Klaper, 2018), particularly of serotonergic system-related genes (Pei et al., 2017; Cunha et al., 2018) and those related to the HPI axis (Kwan et al., 2016). Although acute larval exposure to fluoxetine does not affect thigmotaxis (edge preference), a measure of anxiety-like behavior (Richendrfer et al., 2012), long-term effects of fluoxetine on other behaviors are dependent on the duration of time and specific developmental stage of drug exposure. For example, when zebrafish were exposed to fluoxetine for 24 h starting on 3 days post-fertilization (dpf), only transient effects on motor activity were observed. However, the same fluoxetine treatment initiated on 4 dpf elicited effects on motor activity that persisted up to 14 dpf (Airhart et al., 2007).

Few, if any, studies have examined the impact of developmental SSRI exposure on adult anxiety-like behavior or basal neuroendocrine markers in the zebrafish. Studies in rodents and primates have indicated that serotonin plays a role in the early life programming and function of the HPA axis, possibly by inducing permanent upregulation of the glucocorticoid receptor (Andrews & Matthews, 2004) or the serotonin transporter (Shrestha et al., 2014). It is possible that developmental SSRI exposure in zebrafish may elicit long-term changes in the regulation of the HPI axis as well; however, no studies have examined the impact of chronic exposure to SSRIs during the juvenile period. The juvenile period (approximately 30–89 dpf) occurs after the embryonic and larval periods, but prior to sexual maturation in zebrafish (Kalueff, Stewart & Gerlai, 2014). During the juvenile period, exposure to a mild daily stress plus environmental enrichment decreases anxiety-like behavior in adulthood (DePasquale et al., 2016). However, it is not known whether chronic fluoxetine exposure during the early juvenile period alters the expression of anxiety-like behavior or basal levels of stress hormone responses during adulthood in the zebrafish model.

In the current study, young juvenile zebrafish were chronically exposed to fluoxetine for 14 days and assessed for basal cortisol levels and anxiety-like behavior in adulthood. Anxiety-like behavior was assessed by the novel tank test, a well-validated measure of habituation to novelty. Zebrafish tend to freeze and seek safe zones (e.g., the bottom of the novel tank) when under threat. However, zebrafish also have an innate tendency to explore. Thus, a fish that is demonstrating anxiety-like behavior will be less likely to explore the top of the tank. Based on previously published studies in adult zebrafish, the hypothesis for the current study is that juvenile exposure to fluoxetine will (1) reduce basal cortisol levels and (2) elicit anxiolytic behaviors (e.g., increased exploratory behavior in the top of the novel tank) in adulthood. This study is timely, as the potential impact of juvenile fluoxetine exposure on adult behavior and neuroendocrine responses remains to be fully elucidated.

Materials and Methods

Animals

Wild-type zebrafish (D. rerio) embryos were obtained from the Zebrafish International Resource Center, Eugene, OR. All procedures for raising, feeding, and using zebrafish were carried out by following established recommendations (Westerfield, 2000; Harper & Lawrence, 2011). Larval zebrafish (from hatching until 14 dpf) were maintained in stagnant water at room temperature, fed twice daily with dried, commercially-available larval fish food, and were subject to gentle water exchanges once daily. On 15 dpf, fish were gently moved to the system, a two-shelf, stand-alone housing rack (Aquaneering, San Diego, CA, USA) with a slow drip. The drip was slowly increased every few days to acclimate the fish to a steady stream of water by 30 dpf. The system was maintained on a 14:10 h light/dark cycle, water temperature of 26 ± 2 °C, and pH 7.4 ± 0.2. After 30 dpf, fish were fed once daily with commercially-available flake food.

Juvenile fluoxetine exposure

Starting at 31 dpf, zebrafish (mixed sexes) were randomly moved to one of two 1.8 L housing tanks. Each housing tank was assigned at random to either 0 or 100 μg/L fluoxetine hydrochloride (Santa Cruz Biotechnology, Inc., Dallas, TX, USA). The dose, time, and route of drug administration was based on previously published studies (Egan et al., 2009; Wong, Oxendine & Godwin, 2013; Pittman & Lott, 2014; Pittman & Hylton, 2015). Fluoxetine was prepared as a concentrated stock solution (0.5 mg/ml system water), then, daily, was diluted to the final concentration of 100 μg/L in 1.0 L system water in a clear rectangular polycarbonate container with lid. Fish in the control condition were handled the same as the fluoxetine-treated fish; however, fish were placed in 1.0 L system water in a separate container. The fish were gently transferred by net from their home tank into the appropriate treatment container for 1 h per day for 14 days (modified from Pittman & Hylton, 2015). After the daily treatment, fish were placed back into clean home tanks, which were then placed back in the system. Fresh treatment and control tanks were prepared each day of the chronic treatment. After the treatment, fish were allowed to mature to adulthood in their home tanks without further treatment or handling other than daily maintenance. The sample size of the groups was chosen based on the effect elicited by chronic fluoxetine in adults (Marcon et al., 2016) and was sufficient to give the study >80% statistical power (as per Anderson, Kelley & Maxwell, 2017; https://designingexperiments.com/shiny-r-web-apps/).

Novel tank test

Before the data collection (starting on 90 dpf), both housing tanks were removed from the system and brought into the procedural room and allowed to acclimate for at least 30 min. Individual fish were selected at random, netted, and placed into a trapezoidal novel tank (approximately 3″ × 13″ × 6″, Aquaneering part number ZT180T) for 6 min. The behavior of the fish were recorded and subsequently analyzed with Ethovision XT motion-tracking software (Noldus, Leesburg, VA, USA). Number of entries to the top of tank, time spent in top (sec), distance traveled in the top (cm), latency to enter the top (sec), frequency of freezing, and freezing duration (sec) were used as markers of anxiety-like behavior. A fish that demonstrates anxiety-like behavior will be less likely to explore the top of the tank and will show more freezing behavior. Total distance traveled (cm) and mean speed (cm/sec) were measured as controls to ensure the chronic treatment did not compromise activity levels (Cachat et al., 2010).

Euthanasia and dissection

Fifteen minutes after introduction to the novel tank test, each fish was placed individually in a 50 mL beaker containing 30 mL of 0.1% (100 mg/L) clove oil in system water. Death was determined upon visual examination for cessation of opercular (gill) movement and non-response to tactile stimulation (Davis et al., 2015). The fish were then decapitated; each trunk was placed in a microcentrifuge tube and stored at −20 °C until cortisol extraction and analysis.

Determination of trunk cortisol

Trunk samples were used for assessing levels of cortisol (as per Cachat et al., 2010; Canavello et al., 2011). Briefly, trunk samples were thawed and weighed, then homogenized in 1.0 mL ice-cold 25 mM phosphate-buffered saline (PBS) solution. To extract the cortisol, diethyl ether (5.0 mL) was added to the homogenates. After centrifugation at 4 °C for 15 min at 2,500 rpm, the organic layer containing the cortisol was transferred to a new test tube. This procedure was repeated three times in order to maximize the extracted cortisol from the homogenates. The volatile compounds were allowed to evaporate under a hood. After the evaporation, 1.0 mL 25 mM PBS was added to the lipid layer containing cortisol left in each tube. To determine cortisol levels, a cortisol ELISA was used (Salimetrics, State College, PA, USA) as per the manufacturer’s instructions. Cortisol levels were determined by comparing ELISA binding values to a standard curve. Cortisol values were normalized to trunk weight; the final cortisol data are expressed as ng cortisol/g trunk weight.

Data analysis

Each fish was given a sample number; behavioral and cortisol data were not grouped by treatment until final behavioral measures and cortisol values were obtained. At the end of the experiment, there was a total of n = 17 for the control group and n = 21 for the fluoxetine-treated group. Values were excluded from analyses if two standard deviations above or below the mean for each assay. For the cortisol assay, a total of four samples (two samples from each treatment group) were removed from analyses either due to issues with the extraction procedure (n = 3) or because the value was identified as an outlier in its respective treatment group (n = 1). For the behavioral analysis, several fish (n = 4 in the control group and n = 9 in the fluoxetine-treated group) were removed from analyses due to one or more behavioral parameters falling outside the two standard deviation threshold. Data are presented as group means and the standard errors of the mean. All data were analyzed by independent samples t-tests. JASP software (https://jasp-stats.org/) was used for all statistical analyses. A significance value of p < 0.05 was used as the criterion for a result to reach statistical significance.

Results

Chronic juvenile fluoxetine treatment did not alter trunk cortisol responses

Trunk cortisol levels of zebrafish chronically treated with fluoxetine prior to maturation did not differ from control fish (Fig. 1). An independent samples t-test indicated no significant effect of treatment on trunk cortisol levels (t(23) = −0.027, p = 0.979).

Figure 1 Trunk cortisol levels of young adult zebrafish treated during the juvenile period with and without fluoxetine.

Chronic fluoxetine treatment during the juvenile period (31–44 dpf) did not alter adult levels of trunk cortisol compared to control-treated fish (p = 0.979; independent samples t-test). Values are mean ± SEM of 15–19 fish per group.

Chronic juvenile fluoxetine treatment did not alter motor activity of zebrafish in the novel tank test

Measures of motor activity (total distance moved (cm) and mean speed (cm/sec)) did not differ in zebrafish chronically treated with fluoxetine prior to maturation compared to control fish (Fig. 2). An independent samples t-test indicated no significant effect of treatment on total distance moved (t(23) = −1.053, p = 0.303) and no significant effect of treatment on mean speed (t(23) = −0.540, p = 0.594).

Figure 2 Motor activity measures of young adult zebrafish treated during the juvenile period with and without fluoxetine.

Chronic fluoxetine treatment during the juvenile period (31–44 dpf) did not alter the (A) total distance (p = 0.303; independent samples t-test) or (B) mean speed (p = 0.594; independent samples t-test) of adult fish in the novel tank test compared to control-treated fish. Values are mean ± SEM of 12–13 fish per group.

Chronic juvenile fluoxetine treatment did not alter freezing behavior of zebrafish in the novel tank test

Measures of freezing (number of immobile bouts and total time spent immobile (s)) did not differ in zebrafish chronically treated with fluoxetine prior to maturation compared to control fish (Fig. 3). An independent samples t-test indicated no significant effect of treatment on the number of times zebrafish were immobile (t(23) = 0.482, p = 0.634) and no significant effect of treatment on the time spent immobile (t(23) = −0.539, p = 0.595).

Figure 3 Freezing behaviors of young adult zebrafish treated during the juvenile period with and without fluoxetine.

Chronic fluoxetine treatment during the juvenile period (31–44 dpf) did not alter the (A) number of times immobile (p = 0.634; independent samples t-test) or (B) total time immobile (p = 0.595; independent samples t-test) of adult fish in the novel tank test compared to control-treated fish. Values are mean ± SEM of 12–13 fish per group.

Chronic juvenile fluoxetine treatment did not alter anxiety behavior of zebrafish in the novel tank test

Measures of anxiety-like behavior (distance in top (cm), number of entries to top, total time in top (s), and latency to top (s)) did not differ in zebrafish chronically treated with fluoxetine prior to maturation compared to control fish (Fig. 4). An independent samples t-test indicated no significant effect of treatment on the distance traveled in the top of the novel tank (t(23) = 0.376, p = 0.710), no significant effect of treatment on the number of times zebrafish traversed to the top of the novel tank (t(23) = −0.195, p = 0.847), no significant effect of treatment on the total time spent in the top of the novel tank (t(23) = 0.500, p = 0.622), and no significant effect of treatment on the latency to enter the top (t(23) = 0.020, p = 0.984).

Figure 4 Anxiety-like behaviors of young adult zebrafish treated during the juvenile period with and without fluoxetine.

Chronic fluoxetine treatment during the juvenile period (31–44 dpf) did not alter the (A) distance in top (p = 0.710; independent samples t-test), (B) number of entries to top (p = 0.847; independent samples t-test), (C) time in top (p = 0.622; independent samples t-test), or (D) latency to top (p = 0.984; independent samples t-test) of adult fish in the novel tank test compared to control-treated fish. Values are mean ± SEM of 12–13 fish per group.

Discussion

The present study is the first to investigate the long-term effects of juvenile fluoxetine exposure on adult markers of basal stress regulation in zebrafish. Exposure to fluoxetine for 14 days during the juvenile period (31–44 dpf) was not associated with significant alterations in basal levels of cortisol or indicators of anxiety-like behavior. Thus, the results of the current study suggest that juvenile zebrafish are resilient to or overcome any SSRI-induced neuroadaptations at this dose and time of fluoxetine exposure, at least concerning the basal regulation of the stress response pathway and expression of anxiety-like behavior. These results are consistent with at least one other study in rodents that demonstrated that fluoxetine exposure during adolescence was not associated with increased anxiety-like behavior in adulthood (Norcross et al., 2008). Although some other rodent studies have demonstrated fluoxetine-induced alterations in adult anxiety-like behavior, this could be due, in part, to the timing of the developmental drug exposure. Earlier exposure, such as during the rodent prepubertal period (e.g., around postnatal day 21, as targeted in Ansorge et al., 2004; Oh et al., 2009) may elicit some changes in still-maturing brain pathways that could be more resilient to change slightly later in development. Additional studies done in zebrafish may help to clarify the impact of manipulating serotonin levels at specific developmental stages (e.g., larval, early juvenile, late juvenile) on stress-related markers in adulthood.

Chronic fluoxetine treatment did not affect the long-term regulation of basal levels of stress hormones or behavioral responses during adulthood in the current study. The sample size of the groups in the current study is consistent with other previously published reports that demonstrate that chronic fluoxetine significantly impacts a variety of biological and behavioral markers in adult zebrafish (Maximino et al., 2011; Wong, Oxendine & Godwin, 2013; Giacomini et al., 2016b; Marcon et al., 2016). Additionally, a power analysis indicated that the sample size used in the current study would have been sufficient to detect an equivalent size effect as elicited by chronic fluoxetine exposure in adults. Thus, if there is any effect of developmental exposure to fluoxetine, it has a much smaller impact on basal neuroendocrine responses and anxiety-like behavior compared to adult fluoxetine exposure. Further studies examining fluoxetine exposure in developing vs. adult fish in the same cohort would be necessary to verify the magnitude of differences between these two exposure paradigms.

Additionally, this experiment measured basal levels of stress responses elicited by introduction to a novel tank. The addition of a stronger acute stressor, such as net chasing (De Abreu et al., 2014; Giacomini et al., 2016a, 2016b), may reveal some subtle fluoxetine-induced effects on the expression of stress-induced neuroendocrine and behavioral responses compared to those induced by transfer into a novel tank. Furthermore, in this study, males and females were pooled. Future studies should determine if there are any sex-dependent differences in fluoxetine-induced alterations in stress responses, as other studies have demonstrated sex-dependent differences in stress regulation in response to exposure to chronic unpredictable stress in zebrafish (Rambo et al., 2017) and to fluoxetine exposure in rodents (Pawluski et al., 2012).

Another possible explanation for the results of the current study involve the route of drug administration. Daily handling involved with administering fluoxetine in the current design may have served as a mild stressor and, thus, induced resiliency in the subjects in adulthood. Thus, any fluoxetine-induced alterations may have been overshadowed by the effects of daily handling during the juvenile period. In a previously published study, a mild daily stressor plus environmental enrichment during the juvenile period was associated with decreased anxiety behavior in adulthood (DePasquale et al., 2016). Additional studies would be necessary to determine whether handling alone has an impact on the adult expression of anxiety-like behavior and basal stress hormone responses in zebrafish. Additionally, in this study, young adult zebrafish (approximately 90 dpf) were exposed to zero or 100 μg/L fluoxetine. This dose was chosen based on previous literature using adults (Egan et al., 2009; Wong, Oxendine & Godwin, 2013). Future studies using a range of doses would be necessary to fully elucidate the full impact of juvenile fluoxetine exposure on adult stress-associated behaviors and neuroendocrine responses (Stewart et al., 2014; Sumpter, Donnachie & Johnson, 2014).

Young adult zebrafish in the current study were sacrificed at 15 min post-introduction to the novel tank test in order to assess cortisol levels. This time point was chosen due to previous research demonstrating that the peak cortisol response occurs at least 15 min post-stressor in zebrafish (Ramsay et al., 2009; De Abreu et al., 2014; Tran, Chatterjee & Gerlai, 2014; Pavlidis, Theodoridi & Tsalafouta, 2015). However, the peak cortisol response could have occurred at an earlier or later time point in young adult fish compared to older adult zebrafish. It is possible that young adult zebrafish may have an immature or alternative time course for cortisol release. There is some evidence of possible age-dependent differences in whole-body cortisol at young vs. older adult stages in zebrafish (Ramsay et al., 2009), yet no published studies have established a full cortisol time course of young (90 dpf) adult fish. With regards to the results of the current study, however, the lack of significant basal cortisol alterations in fluoxetine-treated fish compared to controls are paralleled by an absence of altered behavioral responses. Thus, these two different markers support the lack of a long-term effect of juvenile fluoxetine exposure on basal stress-related neuroendocrine responses and anxiety-like behavior.

In sum, juvenile fluoxetine treatment of zebrafish, at the dose and time tested, did not alter adult expression of anxiety-like behavior or basal cortisol responses. This study provides a basis for additional research that is necessary before any conclusions can be drawn regarding permanent neuroadaptive changes elicited by juvenile exposure to fluoxetine. However, the results of the current study suggest that, at least with regards to the basal regulation of stress responses, juvenile fluoxetine exposure at this dose and time does not result in any unexpected or detrimental effects in young adult zebrafish.

Conclusions

In the current study, juvenile zebrafish exposed to fluoxetine for 14 days did not display any alterations in basal cortisol levels or anxiety-like behavior in the novel tank test when assessed in young adulthood. These findings support the notion that SSRI exposure prior to sexual maturation has little effect on the basal regulation of neuroendocrine and behavioral markers of stress in adulthood. This study provides a foundation for additional research on the impact of developmental SSRI exposure on adult markers. A complete understanding would require more extensive resources to fully clarify the impact of dose, targeted developmental window, and differences between basal and stress-induced neuroendocrine and behavioral responses.

Supplemental Information

Supplemental Information 1 Raw behavioral and cortisol data for juvenile fish treated chronically with or without fluoxetine.

The first sheet includes behavioral measures and the second sheet includes cortisol values. All highlighted values were removed before statistical analyses.

Click here for additional data file.

Zebrafish embryos were obtained from the Zebrafish International Resource Center (Eugene, OR). The use of Ethovision XT software was made possible by the Faculty for Undergraduate Neuroscience (FUN) Equipment Loan program. The author thanks Dr. Jonathan Karty of the Indiana University Mass Spectrometry Facility for assistance with some aspects of this study.

Additional Information and Declarations

Competing Interests

Author Contributions

Animal Ethics

Data Availability

The author declares that she has no competing interests.

Maureen L. Petrunich-Rutherford conceived and designed the experiments, performed the experiments, analyzed the data, contributed reagents/materials/analysis tools, prepared figures and/or tables, authored or reviewed drafts of the paper, approved the final draft.

The following information was supplied relating to ethical approvals (i.e., approving body and any reference numbers):

Zebrafish (Danio rerio) are considered to be a species exempt from the Animal Welfare Act (AWA) and, thus, do not need oversight by the Institutional Animal Care and Use Committee (IACUC) at the author’s university if the project is not funded by Public Health Services (PHS).

The following information was supplied regarding data availability:

The raw data are available in a Supplemental File.

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
