# Peer review of "Chronic fluoxetine treatment of juvenile zebrafish (Danio rerio) does not elicit changes in basal cortisol levels and anxiety-like behavior in adulthood"

_PeerJ, doi:10.7717/peerj.6407_

## Round 0.1 · original submission · Major Revisions

The reviewers prepared several comments and queries for your consideration. I believe that all comments will improve your manuscript quality. Regarding reviewers comments, I highlight three points that must be revised in the next version of the manuscript: (1) describe adequately how was done the randomization and blinding of the animals/treatments; (2) there are inconsistencies regarding the number of the animals in each analysis. It is a serious issue. If you removed some outlier for some parameter, it must be removed of the experiment (check the degrees of freedom); (3) the description of the methods should be further expanded to enable replication of the experiments. I, therefore, invite you to revise and resubmit your manuscript, considering the points below as well as the issues raised by the reviewers.

Reviewer 1 ·

Basic reporting

The manuscript would benefit greatly by a thorough edit by a native English speaker.

Experimental design

Review the methodology presenting more details of the procedures.
The sample number presented in the methods is not the same as shown in the figure / table captions.
What is the premise for the removal of outliers data?

Validity of the findings

The data presented are not robust.
The author concludes that "juvenile fluoxetine exposure does not elicit
any adverse or maladaptive effects on stress reactivity in young adult zebrafish, "without presenting any stressful stimulus to conclude on stress reactivity, simply performs an analysis of baseline levels of cortisol.
In addition, many of the references presented by the author (Abreu et al., 2014, 2015, 2017, Giacomini et al., 2016ab, Marcon et al., 2016), fluoxetine alters the stress response only in response to the stressful stimulus (e.g. chasing), fluoxetine has no effect per se.
Thus, the present data do not support the conclusion.

·

Basic reporting

The author investigates an interesting and appealing aspect. The article is written in a clear and understandable way, with a good number of literature references and a sufficient field background /context provided. In addition, the structure is well organized with a sufficient introduction, well explained Material and Methods, an exhaustive chapter of Results and Discussion.

Experimental design

Aim and scope are well defined. However, the following points should be clarified:
1) In experimental design the author did not take the effect of fluoxetine immediately after chronic treatment into account, in terms of cortisol level and behavioural performance. This group is important to verify if Fluoxetine has a biochemical and behavioural effect.
2) In order to validate the data, a further behavioural test on anxiety (light-dark) should be done.
Methods are described with sufficient detail.

Validity of the findings

1) It would be better to express the data concerning the evaluation of the anxious state (in the Table 1) using graphs/figures in order to facilitate the reader.
2) It is not clear the sentence in line 156-157: "Values were excluded from analyses if two standard deviations above or below the mean". How many values did the author exclude?
3) In Discussion (line 194-197) the author can not affirm that his findings are related to serotonin level changes. The author should provide a reference demostrating that fluoxetine alters serontonin levels in zebrafish brain

Additional comments

Line 114: Check the reference order.

·

Basic reporting

The manuscript by Petrunich-Rutherford proposes an investigation of the effect of chronic fluoxetine treatment of juvenile zebrafish in cortisol levels and anxiety-like behavior in adulthood. The paper has a good scientific background and is well-contextualized regarding the literature; in addition, the language used is clear, unambiguous and professional. However, several points are critical to be considered regarding the methodological approaches used (particularly from the point of view of methodology description, experimental design and statistical analysis). I have also raised a series of questions regarding the necessity of further experiments to adequately test the hypothesis of this study.

Abstract and Introduction
1. The goal and the relevance of the study need to be described in a narrower way. What phenomenon does the study intend to model? In the introduction, the author discusses the use of SSRI in adolescent patients suffering from depression and anxiety disorder. However, this study does not use an animal model of depression or anxiety to mimic what is observed in the clinical practice. The author also discusses concepts about stress, but uses no intervention to evaluate a response to a stressor stimulus.

2. There is some important issue of concept. In many instances, the author uses generalist terms assuming that there are changes in brain levels of serotonin, but this parameter was not evaluated in this study. The modulation mechanisms behind the mechanism of action of SSRI are complex and indirectly involve other neurotransmitters and receptors. Therefore, without performing a technique to ascertain the levels of neurotransmitters, such as high-performance liquid chromatography (HPLC), for example, it can not be inferred with certainty that there was actually a change in serotonin levels at the fluoxetine concentration used in this study. (see lines 18 and 195).

3. The author hypothesizes that juvenile exposure to fluoxetine would reduce cortisol levels and elicit anxiolytic effects in the novel tank test in adulthood, but does not propose a neurophysiologic mechanism by which the chronic fluoxetine treatment of juvenile zebrafish would lead to biochemical and behavioral changes in adulthood.

4. Please add the phrase about the novel tank test (100-104) in the Materials and Methods section.

5. The phrase "These data suggest that juvenile fluoxetine exposure does not elicit any adverse or maladaptive effects on stress reactivity in young adult zebrafish" (19-21) is generalist, as the study solely evaluates cortisol levels and behavioral parameters. The term "stress reactivity" may not be ideal, since in this study the animals were not exposed to a stressor stimulus (such as acute or chronic stress).

Materials and methods
6. I believe there is a typographical error in line 147.

Experimental design

The manuscript only partially follows ARRIVE guidelines and there is a need for more minute descriptions of the experimental design. Gerlai, 2018 (Reproducibility and replicability in zebrafish behavioral neuroscience research, doi.org/10.1016/j.pbb.2018.02.005) and Masca et al. (2015; doi:10.7554/eLife.05519) discuss important issues of reproducibility and replicability in scientific research, which must be taken into account:

1. In the Animal section (111-115), besides citing a reference, the author needs to add the maintenance details. In this study animals are used at different stages of development, requiring different types of food, maintenance and housing. How were food (commercial food, artemia or paramecium) and maintenance handled at these different stages? Was the density of animals in the aquariums controlled? What were the parameters of water quality used? Was the system water obtained from reverse osmosis? In addition, it is very important to add the approval number of the ethics committee.

2. Studies show that there are differences between males and females (see Rambo et al. 2017, doi: 10.1016/j.physbeh.2016.12.032), both in behavior and in stress response. Why was this not considered in the statistical analysis?

3. The author states that there are no studies of chronic treatment with fluoxetine in juvenile zebrafish (89-91) and, therefore, there is no known optimal concentration for the drug in this case. Moreover, the concentration used was chosen according to studies performed in adult zebrafish, most of them with acute exposure, not reflecting the design of the present study. In this case, why did the author choose not to establish a dose-response curve of fluoxetine treatment?

6. Most references cited in this manuscript use acute or chronic stress protocols to more concretely evaluate animals' responses to stress. In order to evaluate stress response, I think it would be necessary to expose the animals to a stressor stimulus. In this case, why did not the author use this type of intervention in this study?

7. Please add the dimensions of the apparatus used for the novel tank test (133-134).

8. Please add the P value considered as the cutoff for statistical significance in Data Analysis (155-159).

Validity of the findings

1. An important question is about randomization and blinding. Masca et al. (2015) and Gerlai (2018), examining the problems of reproducibility in biomedical research, state that one of the main reasons for irreproducible research is the lack of blinding/masking. The author describes that randomization was performed to allocate the animals to the treatment groups (control x fluoxetine) (116-117), but were the animals randomized as to the order of testing in the novel tank test? What was the method used for random allocation in the treatment groups? Were experimenters blind to treatment? Were data analysts blind to treatment? These questions need to be addressed and clearly stated in the methods section.

2. Other important issue is that there is a misconduct in the use of degrees of freedom and describing terms of t test statistics. You should not just remove an outlier from a specific parameter; when you consider an outlier animal, you should remove it from all parameters.

3. I believe the term “main effect” should not be used for t test. Please review this concept and change the manuscript.

---

## Round 0.2 · Minor Revisions

The authors did a good job and the article was substantially improved. However, there are some inconsistencies raised by the reviewer who need clarification. I suggest that the authors review these points and respond accordingly. After that I can reevaluate my decision.

·

Basic reporting

See below

Experimental design

See below

Validity of the findings

See below

Additional comments

I reviewed again the manuscript entitled: "Chronic Fluoxetine treatment of juvenile zebrafish (Danio Rerio) does not elicit changes in basal cortisol levels and anxiety-like behavior in adulthood". The rebuttal of the authors is quite satisfactory and I think that the article now is ok for the publication.

·

Basic reporting

When the author uses the term “velocity”, would it be “mean speed”? In this case the correct term must be corrected in the text and figures.

Experimental design

No comment.

Validity of the findings

I am worried that by excluding such a large number of outliers you might lose some important results. How have you evaluated the distribution of your data? Does the data fit a normality curve?
Furthermore, group sizes are different in behavioral analysis and determination of trunk cortisol. When you consider an animal outlier, you should remove it from all tests.

Additional comments

The manuscript by Petrunich-Rutherford proposes an investigation of the effect of chronic fluoxetine treatment of juvenile zebrafish in cortisol levels and anxiety-like behavior in adulthood. The explanations made by the author are valid and the changes improved the manuscript, however, some points still need to be better described (particularly regarding statistical analysis).

---

## Round 0.3 · accepted · Accept

I am pleased to inform you that your manuscript referenced above has been accepted for publication in PeerJ. Congratulations!

#